# Diagnostic accuracy evaluation of a point-of-care antigen test for SARS-CoV-2 and influenza in UK primary care (RAPTOR-C19)

Thomas R. Fanshawe[1], Sharon Tonner [1]*, Philip J. Turner[1], Margaret Głogowska[1], Umasha Ukwatte[1], Cecilia Okusi[1], Jade Cogdale[2], Maria Zambon [3,4], Brian D. Nicholson[1], F. D. Richard Hobbs[1], Gail N. Hayward[1], on behalf of the RAPTOR-C19 Study Group

1 Nuffield Department of Primary Care Health Sciences, University of Oxford, Oxford, United Kingdom, 2 Respiratory Virus Unit, Virus Reference Department, UK Health Security Agency, London, United Kingdom, 3 Influenza and Respiratory Virology & Polio Reference Service, UK Health Security Agency, London, United Kingdom, 4 NIHR Health Protection Research Unit, Imperial College London, London, United Kingdom

◉ These authors contributed equally to this work.
‡ BDN, FDRH and GNH also contributed equally to this work.
¶ Membership of the RAPTOR-C19 study group is provided in the Acknowledgements.
* sharon.tonner@phc.ox.ac.uk

## Abstract

### Objectives

To evaluate the diagnostic accuracy of the Roche SARS-CoV-2 & Flu A/B Rapid Antigen Test at the point of care.

### Design

Prospective diagnostic accuracy study.

### Setting

17 primary care practices in England.

### Participants

500 individuals with symptoms consistent with possible SARS-CoV-2 or influenza infection identified upon presentation to primary care or via medical note review.

### Primary and secondary outcome measures

Sensitivity, specificity and predictive values, compared to a laboratory reference standard of real-time reverse transcription PCR, using samples collected using a combined nasal and oropharyngeal swab.

**Data availability statement:** Data cannot be shared publicly because of participant confidentiality considerations and our institutional responsibility to ensure that any data we collect is used ethically by other researchers in order to maintain public confidence and trust in the research we conduct. All requests for data access are reviewed by Nuffield Department of Primary Care's Primary Care Hosted Research Datasets Independent Scientific Committee (PrimDISC). PrimDISC ensures that data collected in the public interest is used responsibly, ethically and not for spurious reasons. Requests for data access will not be unreasonably refused. Researchers wishing to access data should contact primdisc@phc.ox.ac.uk.

**Funding:** University of Oxford Medical Sciences Division Benefactors Urgent COVID-19 Fund [COVID-19 Research Response Fund Grant 0009325] to BDN, FDRH, JJL, TRF, PJT, GNH, MZ; National Institute for Health and Care Research (NIHR) School for Primary Care Research [SPCR grant 495] to BDN, FDRH, GNH, PJT, JJL, TRF; Urgent Public Health funding received by the CONDOR platform from the NIHR and Asthma + Lung UK [NIHR UPH grant COV0051] to BDN, PJT, GNH. PT, TRF, MG, UU, FDRH and GNH have received funding from the National Institute for Health and Care Research (NIHR) Community Healthcare MedTech and In Vitro Diagnostics Co-operative (MIC) (MIC 2016-018). PJT, TRF, MG, UU, BN and GNH currently receive funding from the NIHR HealthTech Research Centre (HRC) in Community Healthcare (NIHR205287) at Oxford Health NHS Foundation Trust. TRF and FDRH receive funding from the NIHR Applied Research Collaboration Oxford and Thames Valley at Oxford Health NHS Foundation Trust (NIHR200172). The views expressed are those of the authors and not necessarily those of the NHS, the NIHR or the Department of Health and Social Care. Roche International Ltd (R69261/CN085) provided consumables and site training free of charge, together with a grant to the University of Oxford to support the study. The funders had no role in study design, data collection and analysis, decision to publish, or preparation of the manuscript.

**Competing interests:** We have read the journal's policy and the authors of this manuscript

## Results

Of 481 participants with available index and reference test results, 5.6% (27/481) were reference standard-positive for SARS-CoV-2, 11.4% (55/481) for Influenza A and 1.9% (9/481) for Influenza B. The sensitivity of the antigen test to detect SARS-CoV-2 was 70.4% (19/27, 95% CI 49.6–86.2%) and specificity was 99.3% (451/454, 95%CI 98.1–99.9%). For Influenza A, sensitivity was 29.1% (16/55, 95% CI 17.6–42.9%) and specificity 98.6% (420/426, 97.0–99.5%), and for Influenza B, sensitivity was 22.2% (2/9, 2.8–60.0%) and specificity 98.1% (463/472, 96.4–99.1%).

## Conclusions

In a primary care population of symptomatic individuals, the assay was highly specific and had moderate sensitivity to detect SARS-CoV-2, but did not detect the majority of influenza infections.

## Registration

ISRCTN14226970.

## Introduction

The emergence of SARS-CoV-2 in 2019 and subsequent pandemic led to the urgent need to develop not only effective treatments and vaccinations but also diagnostic tests. Diagnostic tests are especially valuable in pandemic settings, as at the outset of any pandemic, diagnostics are among the few tools that public health agencies have at their disposal to control infection before the availability of therapies and vaccines and continue to be used after the initial pandemic phase to monitor public health [1].

While polymerase chain reaction (PCR) tests became rapidly available at the outset of the COVID-19 pandemic, the turnaround time of these laboratory-based assays meant that rapid point-of-care tests (POCT) were still required for use in the community [2]. This was necessary to promptly identify infected individuals and disrupt community transmission, to shield clinically vulnerable individuals and groups, to guide access to treatments as they became available, and to reduce the burden on laboratory services and hospitals. Multiplex diagnostic technologies have encouraging potential if they can accurately distinguish between infectious disease pathogens in individuals who present with similar clinical signs and symptoms [3].

The RAPid community Test evaluations fOR COVID-19 (RAPTOR-C19) study was established in 2020 to evaluate the performance of POCTs for SARS-CoV-2 in community healthcare settings in the UK [4,5]. From 2022, RAPTOR-C19 expanded to include evaluations of multiplex rapid antigen tests as the phasing out of social distancing regulations saw the re-emergence of seasonal influenza. COVID-19 and influenza can be difficult to distinguish from each other based on clinical symptoms alone [6]. In this fourth arm of the study, we evaluated the Roche-branded SD Biosensor SARS-CoV-2 & Flu A/B Rapid Antigen Test.

have the following competing interests: The authorship declares funding support for this study from the University of Oxford Medical Sciences Division Benefactors Urgent COVID-19 Fund, the National Institute for Health and Care (NIHR) School of Primary Care Research, and Urgent Public Health funding for the CONDOR Platform from the NIHR and Asthma+Lung UK. Roche International Ltd provided consumables and site training free of charge, together with a grant to the University of Oxford to support the study. TRF declares NIHR support from the NIHR Community Healthcare MIC for diagnostic evaluation research. PJT declares support from the NIHR Community Healthcare MIC for diagnostic evaluation research. PJT has provided expert support to the Longitude Prize AMR competition administration which is unrelated to this project and for which the University of Oxford received an honorarium. MZ declares her unpaid activities as the Chair of the charitable organisation ISIRV and her membership of the UK SAGE, NERVTAG and JCVI groups. This does not alter our adherence to PLOS ONE policies on sharing data and materials.

## Methods

### Design

RAPTOR-C19 is a prospective observational parallel diagnostic accuracy platform. In the fourth arm of the study, individuals were recruited on presentation to participating community health care settings in England with symptoms of viral respiratory illness consistent with potential SARS-CoV-2 infection [7]. Adults and children were eligible to participate. Exclusion criteria were inability to consent (adult only), requiring immediate hospitalisation, and having previously participated in the current evaluation.

Further details of the overall design of RAPTOR-C19 can be found in the published protocol [8]. However, following approved protocol amendments, participant-reported follow-up data was not collected for this arm of the study and serology collection was not performed.

### Ethical approval

RAPTOR-C19 was approved by the North-West Liverpool Central Research Ethics Committee, reference 20/NW/0282. Participants, or a parent/legal guardian if the participant was aged 15 or under, provided verbal informed consent before any study procedures or data collection commenced, which was subsequently recorded in an electronic case report form (eCRF). An electronic participant information leaflet, with age-appropriate versions available for those aged 15 and under, was provided prior to consent.

### Recruitment

A total of 500 participants were recruited from 17 primary care practices in England across the former Thames Valley & South Midlands, East of England, North West London, South London, West of England, East Midlands, and Yorkshire & Humber Clinical Research Network (CRN) regions. Participants were invited to join the study either opportunistically on presentation or via medical note review of recent presentations with symptoms. Recruitment to the Roche arm commenced on 19th January 2023 but was paused on 29th March 2023 due to low detection rates of SARS-CoV-2 in the recruited cohort. Recruitment restarted on 4th December 2023 and continued until 15th March 2024 in line with the typical viral respiratory 'winter bug' season in the Northern Hemisphere. During the January-March 2023 phase of recruitment, Omicron, GRA XBB.1.5 and GRA CH.1.1 variants of SARS-CoV-2 were found to be in circulation, while in the November 2023-March 2024 season GRA JN.1 ("Pirola") predominantly circulated. In the 2023/24 flu season the majority of circulating virus was H1N1 with around 30–40% H3N2 [9,10].

### Baseline measurements

Participants' baseline visit data was collected via an eCRF (uMed, Warner Yard, London, UK January-March 2023 and REDCap V14.0.12, Nashville, USA from November 2023 to study close [11,12]) and included information on symptoms and their duration, SARS-CoV-2 vaccination status and demographic information. Results

of the index test and any issues experienced were also recorded on the eCRF alongside confirmation that a reference standard test had been obtained.

### Index test

The POCT evaluated in this arm of the study is the Roche-branded and distributed SD Biosensor SARS-CoV-2 & Flu A/B Rapid Antigen Test (REF 9901-CVFL-01C, Roche Diagnostics GmbH). This test is for healthcare professional use only. It is a qualitative, user-interpreted, disposable lateral flow test which multiplexes the identification of viral antigens of both SARS-CoV-2 and influenza A/B from a common patient sample [13]. Each assay incorporates an internal quality control that resolves as a visible line to indicate a valid test. Test kits of 25 assays included both positive and negative external quality control materials for SARS-CoV-2 and influenza A/B. Sites were instructed to confirm the integrity of new kits as they were delivered using these materials before proceeding to participant testing.

Unilateral nasopharyngeal samples were collected from participants using swabs provided as an element of the test kit, in accordance with the manufacturer's documentation. The evaluation incorporated a single manufacturer 'LOT' (batch) of assays. All staff carrying out study procedures completed training in the use of the test via a training video overseen by Roche representatives who were available to address queries prior to sites commencing recruitment. A quick reference guide and instructions for use (IFU) were also provided and were required to be followed. Recruiting staff who performed testing were asked to report test ease-of-use on a five-point scale (ranging from 'not very easy' to 'very easy'). The index test result was not shared with patients.

### Reference standard

Samples for reference standard testing were collected from nasal and oropharyngeal sites using a combined swab prior to storage and transport in viral transport medium (REF MW012, Dryswab™; REF MW950T50, Sigma Virocult® tube and medium, Medical Wire & Equipment Co. (Bath) Ltd, Corsham, Wiltshire, UK) contemporaneous to the collection of samples for index testing. SARS-CoV-2 and influenza A and B were detected by RT-PCR (real-time reverse transcription PCR), with the following targets: ORF1ab and E gene regions of SARS-CoV-2; matrix (M) and haemagglutinin (HA) genes of influenza A; nucleoprotein (NP) gene of influenza B. Reference standard equipment and methodology were as previously described [5] with primer and probe combinations listed in S1 Table.

Positive or negative results were reported for SARS-CoV-2, influenza A (inclusive of subtype), and influenza B. Results for SARS-CoV-2, influenza A subtypes and influenza B were accompanied by RT-PCR cycle threshold values (Ct) for each molecular target. No information about alternative infections in those testing negative for SARS-CoV-2 and influenza was accessible to the study team.

Reference samples were processed at the UK Health Security Agency (UKHSA) Respiratory Virus Reference laboratory (UKHSA Virus Reference Department (VRD), Colindale, London, UK). Staff who performed the reference standard were unaware of index test results. Results of the reference standard test were shared with the general practice through standard UK HSA reporting mechanisms and it was permitted to share these results with the patient.

### Sample size

We planned to recruit a sufficient number of participants to obtain 150 reference standard positive cases of SARS-CoV-2, as per the Target Product Profile for point-of-care SARS-CoV-2 detection tests [14]. For a POCT with at least 90% sensitivity, we would expect to have a standard error of the sensitivity estimate of ≤2.5%, provided the prevalence of SARS-CoV-2 were 30% or higher. In the event, the prevalence of SARS-CoV-2 throughout the recruitment period was much lower than anticipated, and lower than had been observed in the previous arms of the study (approximately 5%, compared to 39% in October 2020-October 2021 and 17% in June-December 2022) [5]. As continuation to the original target number

of positive cases would have made the total required recruitment prohibitive, the study was terminated without interim analysis after 500 participants had been recruited.

## Statistical analysis

We calculated the proportion of RT-PCR results that were positive for SARS-CoV-2, and the sensitivity, specificity, and predictive values of the POCT, with exact 95% confidence intervals (95% CIs). The number of individuals with missing index test and/or reference standard results were reported but these individuals were excluded from the analysis. We also calculated diagnostic performance in relation to the RT-PCR cycle threshold (Ct). Numbers of true positive, true negative, false positive and false negative results are also presented via pre-specified subgroups split by age, sex, ethnicity, baseline symptoms and time since symptom onset. A subgroup analysis was additionally performed by CRN region. For the secondary outcomes (influenza A and B) we used similar methods to summarise diagnostic accuracy performance, additionally subdividing influenza A cases by Haemagglutinin (HA) subtype. Statistical analysis was performed using RStudio 2022.02.3 including the epiR package [15].

## Ethnography methodology

Ethnographic observation took place in February-March 2024 at two participating primary care practices (one in the Thames Valley & South Midlands CRN region and one in the Yorkshire & Humber CRN region). Observation was carried out by two researchers (MG and UU) who each observed in one practice. Three health care professionals who were responsible for conducting the test in their practices were observed (two General Practitioners (GPs) and one Research Nurse).

Observations were informed by a list of sensitising topics compiled from a schedule utilised in previous ethnographic studies of POCT [16]. In each clinic, verbal consent from patients for the researcher to be present was obtained prior to observation. Risk assessment was carried out for the researcher visits and appropriate personal protective equipment worn. During the observations the researchers wrote field notes. With the health care professionals permission, audio-recording was used to capture informal conversations with them about the testing process which occurred during the visits. Topics discussed included training received and support available to health care professionals who were using the test, how quality control took place, the management and tracking of equipment and consumables, as well as more general issues about conducting testing in community settings. The notes and supplemental audio recordings formed the basis of the analysis.

Analysis began with reading and becoming familiar with the field notes of the observations and conversations, noting and recording initial issues and themes and then conducting systematic coding alongside the text of the field notes. The codes were then grouped into categories, which included aspects and characteristics of the Roche test and the testing process. Key considerations during the analysis were the usability of the testing equipment in the context observed, the acceptability of the testing process to health care professionals and patients, and how use of the test fitted into the workflow of the clinician.

## Results

### Participant characteristics

Of 500 participants who consented to recruitment to the study, seven were excluded as no participant information was collected following recruitment, a further five had no available POCT result (one of which was explicitly recorded by the device as 'no result', and the other four were unrecorded), and a further seven had the POCT performed but no laboratory PCR result was available (Fig 1). The primary analysis used data from the 481 participants who had both POCT and PCR results available.

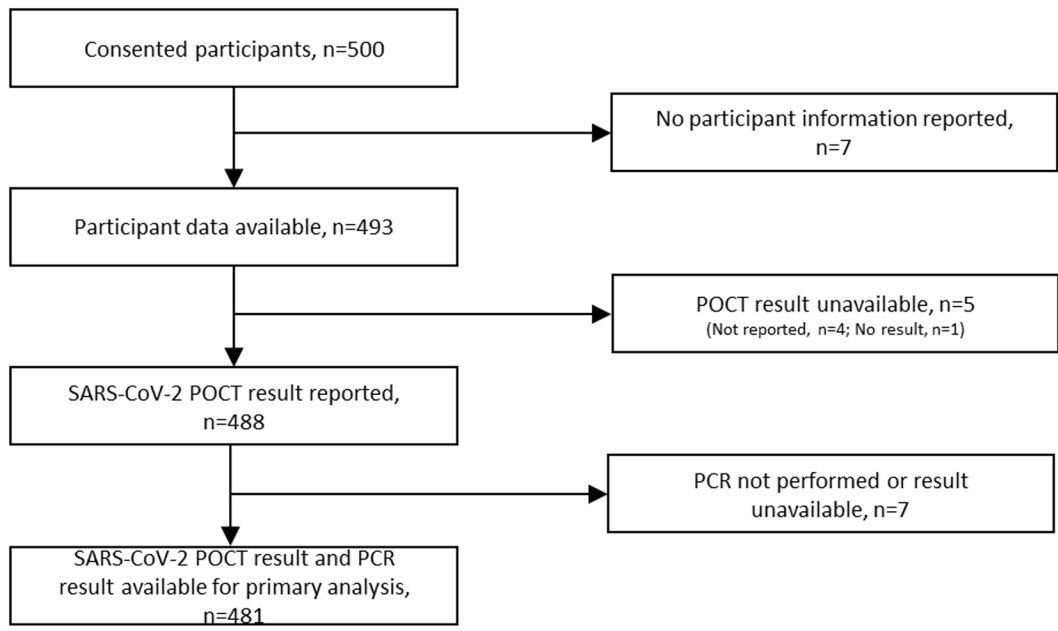

**Fig 1. Study flowchart.**

Table 1 shows participant characteristics, split by whether positive/negative for SARS-CoV-2 in their PCR result. In the primary analysis sample, 41% (198/481) were male and average age was 28 years, with 36% (175/481) aged under 16 and the majority of the children having been recruited from a single site. Average time since self-reported symptom onset was 3.7 days. Most participants reported onset of first symptoms between two and five days before recruitment (Fig 2) and the most prevalent symptoms were cough (81%, 388/481), fever (49%, 234/481) and sore throat (42%, 200/481).

## Diagnostic accuracy

In the primary analysis sample, 5.6% of participants had a positive SARS-CoV-2 PCR result (27/481, 95% CI 3.7 to 8.1%), 11.4% were positive for Influenza A (55/481, 8.7 to 14.6%) and 1.9% were positive for Influenza B (9/481, 0.9 to 3.5%). Of the 55 samples that were positive for Influenza A, 29 were subtype H1N1, 22 were subtype H3N2 and subtyping was not recorded for the remaining four.

Diagnostic performance of the POCT is summarised in Table 2. Sensitivity to detect SARS-CoV-2 was 70.4% (19/27, 49.8 to 86.2%) and specificity 99.3% (451/454, 98.1 to 99.9%). For Influenza A, sensitivity was 29.1% (16/55, 17.6 to 42.9%) and specificity 98.6% (420/426, 97.0 to 99.5%). For Influenza B, sensitivity was 22.2% (2/9, 2.8 to 60.0%) and specificity 98.1% (463/472, 96.4 to 99.1%).

Information about diagnostic performance by pre-specified subgroups of participant and stratified further by CRN site are shown in S1 File. No diagnostic accuracy estimates are shown for these subgroups as the number of positive samples was lower than anticipated. The POCT correctly detected as positive most of the PCR-positive samples that had lower Ct values, but not those at higher Ct values (Table 3).

## Ease of use

In the primary analysis sample, a problem with performance of the POCT was logged for four participants (0.8%) but a POCT result was recorded for all of these, possibly after repeating the test. On a five-point scale, ease-of-use was recorded by the

**Table 1. Baseline characteristics (reported as number (%) or mean (standard deviation)).**

| | All partici-pants (n = 493) | Participants included in primary analysis (n = 481) | Participants with positive SARS-CoV-2 PCR result (n = 27) | Participants with negative SARS-CoV-2 PCR result (n = 454) |
|---|---|---|---|---|
| Male sex | 204 (41%) | 198 (41%) | 10 (37%) | 188 (41%) |
| Age (years) | 28 (22) | 28 (22) | 36 (22) | 28 (22) |
| - < 16 | 180 (37%) | 175 (36%) | 6 (22%) | 169 (37%) |
| − 16-39 | 164 (33%) | 161 (33%) | 9 (33%) | 152 (33%) |
| − 40-59 | 104 (21%) | 101 (21%) | 8 (30%) | 93 (20%) |
| − 60 + | 45 (9%) | 44 (9%) | 4 (15%) | 40 (9%) |
| Ethnicity | | | | |
| - White | 343 (70%) | 335 (70%) | 20 (74%) | 315 (69%) |
| - Asian | 91 (18%) | 87 (18%) | 4 (15%) | 83 (18%) |
| - Black | 22 (4%) | 22 (5%) | 2 (7%) | 20 (4%) |
| - Mixed-White and Black African/ Caribbean | 2 (<1%) | 2 (<1%) | – | 2 (<1%) |
| - Mixed-White and Asian | 2 (<1%) | 2 (<1%) | – | 2 (<1%) |
| - Mixed-Other | 11 (2%) | 11 (2%) | 1 (4%) | 10 (2%) |
| - Other | 22 (4%) | 22 (5%) | – | 22 (5%) |
| Time since first symptom | | | | |
| - First symptom within previous 14 days | 486 (99%) | 475 (99%) | 27 (100%) | 448 (99%) |
| - Number of days since first symptom* | 3.7 (2.2) | 3.7 (2.2) | 3.1 (1.5) | 3.8 (2.3) |
| New symptoms within previous 14 days† | | | | |
| - Any symptom | 491 (>99%) | 480 (>99%) | 27 (100%) | 453 (>99%) |
| - Fever | 241 (49%) | 234 (49%) | 17 (63%) | 217 (48%) |
| - Cough | 397 (81%) | 388 (81%) | 22 (81%) | 366 (81%) |
| - Fatigue | 119 (24%) | 118 (25%) | 9 (33%) | 109 (24%) |
| - Shortness of breath | 119 (24%) | 118 (25%) | 7 (26%) | 111 (24%) |
| - Sputum | 114 (23%) | 111 (23%) | 6 (22%) | 105 (23%) |
| - Loss of smell or change in taste | 42 (9%) | 42 (9%) | 6 (22%) | 36 (8%) |
| - Muscle ache | 100 (20%) | 100 (21%) | 6 (22%) | 94 (21%) |
| - Chills | 73 (15%) | 73 (15%) | 6 (22%) | 67 (15%) |
| - Dizziness | 42 (9%) | 42 (9%) | 1 (4%) | 41 (9%) |
| - Headache | 108 (22%) | 107 (22%) | 8 (30%) | 99 (22%) |
| - Sore throat | 202 (41%) | 200 (42%) | 15 (56%) | 185 (41%) |
| - Hoarseness | 69 (14%) | 68 (14%) | 5 (19%) | 63 (14%) |
| - Nausea or vomiting | 35 (7%) | 35 (7%) | 1 (4%) | 34 (7%) |
| - Diarrhoea | 16 (3%) | 16 (3%) | 1 (4%) | 15 (3%) |
| - Nasal congestion | 127 (26%) | 126 (26%) | 10 (37%) | 116 (26%) |
| - Other | 47 (10%) | 47 (10%) | 2 (7%) | 45 (10%) |

*Calculated among participants whose reported date of first symptom fell up to 14 days before the recruitment date.

†Symptoms that, by participant self-report, began within 14 days before the recruitment date.

health care professionals performing the test as point 1 (not very easy) on 4 occasions (0.8%), point 2 on 21 occasions (4%), point 3 on 55 occasions (11%), point 4 on 75 occasions (16%) and point 5 (very easy) on 326 occasions (68%).

No adverse events as a result of performing the POCT were reported.

## Qualitative findings

Overall, RAPTOR-C19 site staff found the testing kits usable in the settings where they were collecting and processing samples, fitting into their workflow. They felt comfortable using a lateral flow device, which had become very familiar during the pandemic. The staff were enthusiastic about the test being able to process samples, not just for SARS-CoV-2

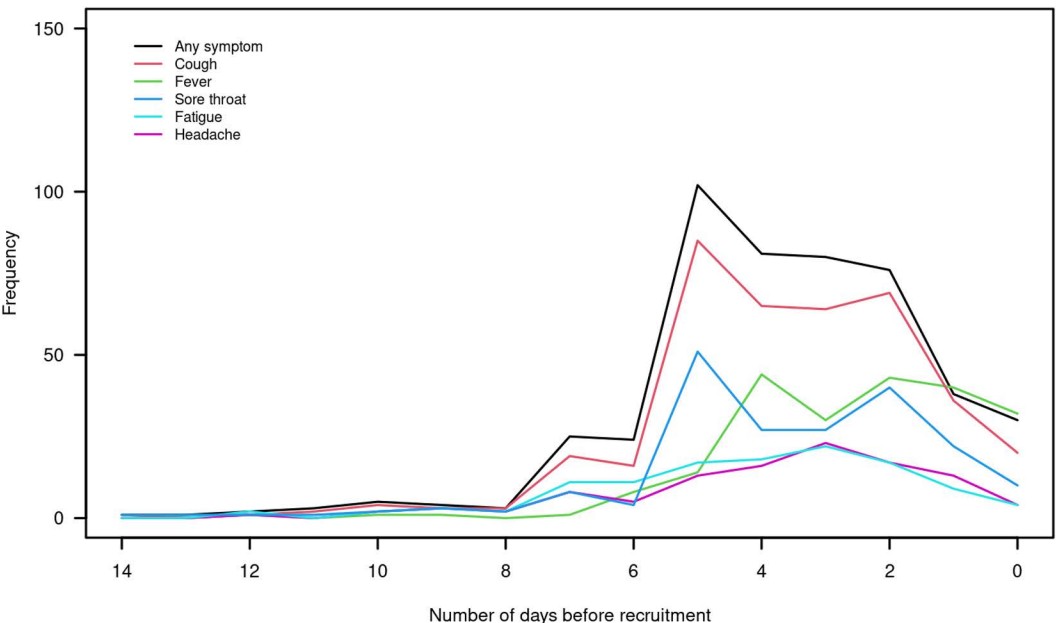

**Fig 2. Number of days prior to recruitment (day 0) of first onset of any symptom; cough; fever; sore throat; fatigue; headache, of recruited participants.**

**Table 2. Diagnostic accuracy.**

| | | Reference standard result | | | | Sensitivity | Specificity | PPV | NPV |
|---|---|---|---|---|---|---|---|---|---|
| | | Pos-itive | Neg-ative | Total reported | Not reported | | | | |
| POCT SARS-CoV-2 result | Positive | 19 | 3 | 22 | 0 | 70.4% (49.8%, 86.2%) | 99.3% (98.1%, 99.9%) | 86.4% (65.1%, 97.1%) | 98.3% (96.6%, 99.2%) |
| | Negative | 8 | 451 | 459 | 7 | | | | |
| | Total | 27 | 454 | 481 | 7 | | | | |
| POCT Influenza A result | Positive | 16 | 6 | 22 | 2 | 29.1% (17.6%, 42.9%) | 98.6% (97.0%, 99.5%) | 72.7% (49.8%, 89.3%) | 91.5% (88.6%, 93.9%) |
| | Negative | 39 | 420 | 459 | 5 | | | | |
| | Total | 55 | 426 | 481 | 7 | | | | |
| POCT Influenza B result | Positive | 2 | 9 | 11 | 0 | 22.2% (2.8%, 60.0%) | 98.1% (96.4%, 99.1%) | 18.2% (2.3%, 51.8%) | 98.5% (97.0%, 99.4%) |
| | Negative | 7 | 463 | 470 | 7 | | | | |
| | Total | 9 | 472 | 481 | 7 | | | | |

(Influenza A subtypes: Subtype H1: 12/29 correctly diagnosed (sensitivity 41.4% (23.5%, 61.1%)); Subtype H3: 3/22 correctly diagnosed (13.6% (2.9%, 34.9%)); No subtype reported: 4 Influenza A cases).

but also for Influenza A & B from the same swab. They felt it was useful at times when a number of respiratory viruses were in circulation and particularly when many of the patients attending the GP practices had suspected influenza due to the presence of muscle aches, which was perceived as a common flu symptom

The staff reported that the quality control testing was straightforward. They found the processing of the test acceptable and they considered the 15–20 minute timeframe for the result to appear satisfactory. However, the biggest challenge the staff encountered was the nature of the necessary swabbing. Nasopharyngeal swabbing was considered quite difficult to achieve. Staff expressed doubt as to whether they were able to successfully reach the nasopharynx and keep the swab in

**Table 3. Diagnostic performance by Ct value.**

**SARS-CoV-2**

| | | ORF1ab | | | | E gene | | | |
|---|---|---|---|---|---|---|---|---|---|
| | Ct value | < 20 | 20-25 | 25-30 | > 30 | < 20 | 20-25 | 25-30 | > 30 |
| POCT result | Positive | 5 | 12 | 2 | 0 | 5 | 11 | 3 | 0 |
| | Negative | 0 | 1 | 3 | 4 | 0 | 1 | 3 | 4 |

**Influenza**

| | | Influenza A | | | | Influenza B | | | |
|---|---|---|---|---|---|---|---|---|---|
| | Ct value | < 20 | 20-25 | 25-30 | > 30 | < 20 | 20-25 | 25-30 | > 30 |
| POCT result | Positive | 1 | 3 | 7 | 4 | 0 | 2 | 0 | 0 |
| | Negative | 1 | 7 | 14 | 11 | 0 | 2 | 3 | 2 |

(Seven Influenza A participants had missing Ct values).

place for the 10 seconds required by the manufacturer's IFU. Observations across the sites and different operators also showed that patients did not easily tolerate this form of swabbing. One health care professional commented that the need for nasopharyngeal swabbing as part of this test meant that they would not be comfortable using it with children.

## Discussion

### Summary of findings

This evaluation of the Roche-branded SD Biosensor SARS-CoV-2 & Flu A/B Rapid Antigen Test was conducted as a pragmatic study but was limited by low prevalence of SARS-CoV-2 during the period of recruitment. This highlights that evaluating new diagnostics that are needed in pandemics remains challenging outside such periods due to limited presentations, and that infrastructure to quickly implement diagnostic evaluations should be part of any pandemic preparedness plans.

For the POCT under evaluation, the 70.4% sensitivity to detect SARS-CoV-2 falls below the UK government's Target Product Profile threshold for 'acceptable' at 80% sensitivity and well below the 'desirable' threshold of 97% sensitivity, although there was considerable uncertainty in the estimate [14]. There is no available Target Product Profile for influenza POCT's, but the RAPP-ID consortium for LRTIs recommends minimum 85% sensitivity [17] and current performance estimate of 29.1% sensitivity for influenza A and 22.2% sensitivity for influenza B lies well below this, even at lower Ct, and despite the target sample size not having been reached. Performance for influenza is towards the lower end of estimates from previous meta-analyses of other rapid devices [18].

### Strengths and limitations

A major strength is that we conducted a large pragmatic study in the setting where these tests should be used, therefore filling an evidence gap for POCT performance at the point of care. A major limitation of the study was the low observed prevalence of SARS-CoV-2, which caused recruitment to be paused at the end of the first winter season. Although this is a consequence of conducting a study in a real-world setting and may reflect fewer symptomatic patients attending primary care, prevalence was much lower than had been observed in similar studies between 2020 and 2022 [4,5] and resulted in wide confidence intervals for some of the diagnostic performance measures.

There are also other limitations. While the use of a single 'lot' of assays aided in the logistics of study delivery, our evaluation lacks the clinical reality of multiple deployed lots. Results relate only to the predominant viral strains during the study period and locations and cannot necessarily be extrapolated to future strains. Our findings of poor sensitivity for influenza may be partly due to the need for nasopharyngeal swabbing when performing the POCT. While nasopharyngeal swabs are more sensitive than anterior nasal swabs [19,20], they are not generally as well tolerated [21], as supported by our ethnographic

observations. In this study, it was not possible to tell if a nasopharyngeal swab was successfully obtained for each participant, and the ethnographic findings indicated that some users had low confidence in using nasopharyngeal swabs correctly. The reference standard used a combined nasal and oropharyngeal swab; we think it unlikely that this is a major source of diagnostic error as it has been shown in a systematic review to have similar performance to a nasopharyngeal swab for RT-PCR [22].

## Comparison with other studies

There are few previous evaluations in primary care settings of the POCT considered in this study. The manufacturer-reported performance of this assay in symptomatic patients is 95.25% sensitivity (Ct<=30) and 99.35% specificity for SARS-CoV-2; 100% sensitivity (Ct<=30) and 99.75% specificity for Influenza A; and "using retrospective specimens", 100% sensitivity and 100% specificity for Influenza B [13]. A laboratory evaluation of a Roche-branded SARS-CoV-2/Flu A and Flu B combination Rapid Antigen Test in Australia in 2022 reported similarly good performance for lower Ct values (<=26) but failed to detect SARS-CoV-2 variant BA.2 at Ct>=29.5 [23].

Viral loads following influenza infection typically peak within a few days after initial infection and at similar time as the first clinical symptoms [24]. SARS-CoV-2 may have a longer incubation period, with viral load peaking several days after symptom onset and longer duration of viral shedding [25,26]. As patients entered this study on average 3.7 days after first symptom onset, the timelines may have been more closely aligned to detection of SARS-CoV-2 than influenza at the point of care, especially if some nasopharyngeal swabbing was performed suboptimally. Nevertheless, our study represents a scenario in which the POCT might typically be deployed.

## Conclusion

The assay assessed was highly specific and had moderate sensitivity to detect SARS-CoV-2, but did not detect the majority of influenza infections. Negative POCT results should not be used to rule out the possibility of infection. Further work is needed to ascertain, outside a pandemic setting, the best way to integrate POCTs which offer high specificity and moderate sensitivity into care pathways for acute respiratory infections in the community.

## Supporting information

**S1 Table. Primers and probes for the detection of Influenza A & B viruses by rtRTPCR.**
(DOCX)

**S1 File. Subgroup analysis tables.**
(DOCX)

## Acknowledgments

We are grateful to all study participants, RAPTOR-C19 site staff, and the staff of the NIHR Clinical Research Network, Thames Valley and South Midlands for their support for the study. We would like to thank Micheal McKenna[1] for RedCAP support, patients and practices in the Oxford-Royal College of General Practitioners Research and Surveillance Centre (RSC) who share pseudonymised data to support research and surveillance (UKHSA is the principal sponsor of the RSC); EMIS, TPP, Vision and Wellbeing for assistance with pseudonymised data extraction.

Membership of the RAPTOR-C19 Study Group is as follows: Katie Arundell[1], Gail N. Hayward[1], F. D. Richard Hobbs[1], Heather Kenyon[2], Joseph J. Lee[1], Kathryn Lucas[2], Brian D. Nicholson[1], Jessica Smylie[1], Sharon Tonner[1]* , Philip J. Turner[1].

1 Nuffield Department of Primary Care Health Sciences, University of Oxford, UK

2 NIHR Clinical Research Network, Thames Valley and South Midlands, Oxford, UK

*lead and corresponding author for the RAPTOR-C19 Study Group – sharon.tonner@phc.ox.ac.uk

## Author contributions

**Conceptualization:** Thomas R. Fanshawe, Philip J. Turner, Maria Zambon, Brian D. Nicholson, F. D. Richard Hobbs, Gail N. Hayward.

**Data curation:** Sharon Tonner, Thomas R. Fanshawe, Philip J. Turner, Margaret Głogowska, Umasha Ukwatte, Cecilia Okusi, Jade Cogdale.

**Formal analysis:** Thomas R. Fanshawe, Margaret Głogowska.

**Funding acquisition:** Sharon Tonner, Thomas R. Fanshawe, Philip J. Turner, Brian D. Nicholson, F. D. Richard Hobbs, Gail N. Hayward.

**Investigation:** Margaret Głogowska, Umasha Ukwatte.

**Methodology:** Philip J. Turner.

**Project administration:** Sharon Tonner.

**Resources:** Maria Zambon.

**Supervision:** F. D. Richard Hobbs, Gail N. Hayward.

**Writing – original draft:** Sharon Tonner, Thomas R. Fanshawe, Philip J. Turner.

**Writing – review & editing:** Sharon Tonner, Thomas R. Fanshawe, Philip J. Turner, Margaret Głogowska, Umasha Ukwatte, Cecilia Okusi, Maria Zambon, Jade Cogdale, Brian D. Nicholson, F. D. Richard Hobbs, Gail N. Hayward.

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
