## [Decision Letter · Decision Letter 0]

25 Mar 2025

PONE-D-24-60482Diagnostic accuracy evaluation of a point-of-care antigen test for SARS-CoV-2 and influenza in UK primary care (RAPTOR-C19)PLOS ONE

Dear Dr. Tonner,

Thank you for submitting your manuscript to PLOS ONE. After careful consideration, we feel that it has merit but does not fully meet PLOS ONE’s publication criteria as it currently stands. Therefore, we invite you to submit a revised version of the manuscript that addresses the points raised during the review process.

We look forward to receiving your revised manuscript.

Kind regards,

Bushra Akhtar, Ph.D

Academic Editor

PLOS ONE

**Journal Requirements:**

1. When submitting your revision, we need you to address these additional requirements. Please ensure that your manuscript meets PLOS ONE's style requirements, including those for file naming. The PLOS ONE style templates can be found at https://journals.plos.org/plosone/s/file?id=wjVg/PLOSOne_formatting_sample_main_body.pdf and https://journals.plos.org/plosone/s/file?id=ba62/PLOSOne_formatting_sample_title_authors_affiliations.pdf 2. We note that the grant information you provided in the ‘Funding Information’ and ‘Financial Disclosure’ sections do not match.  When you resubmit, please ensure that you provide the correct grant numbers for the awards you received for your study in the ‘Funding Information’ section. 3. Thank you for stating the following financial disclosure: University of Oxford Medical Sciences Division Benefactors Urgent COVID-19 Fund [COVID-19 Research Response Fund Grant 0009325] to BDN, FDRH, JJL, TRF, PJT, GNH, MZ; National Institute for Health and Care Research (NIHR) School for Primary Care Research [SPCR grant 495] to BDN, FDRH, GNH, PJT, JJL, TRF; Urgent Public Health funding received by the CONDOR platform from the NIHR and Asthma + Lung UK [NIHR UPH grant COV0051] to BDN, PJT, GNH. PT, TRF, MG, UU, FDRH and GNH have received funding from the National Institute for Health and Care Research (NIHR) Community Healthcare MedTech and In Vitro Diagnostics Co-operative (MIC) (MIC 2016-018). PJT, TRF, MG, UU, BN and GNH currently receive funding from the NIHR HealthTech Research Centre (HRC) in Community Healthcare (NIHR205287) at Oxford Health NHS Foundation Trust. TRF and FDRH receive funding from the NIHR Applied Research Collaboration Oxford and Thames Valley at Oxford Health NHS Foundation Trust (NIHR200172). The views expressed are those of the authors and not necessarily those of the NHS, the NIHR or the Department of Health and Social Care. Roche International Ltd provided consumables and site training free of charge, together with a grant to the University of Oxford to support the study.   Please state what role the funders took in the study.  If the funders had no role, please state: "The funders had no role in study design, data collection and analysis, decision to publish, or preparation of the manuscript." If this statement is not correct you must amend it as needed. Please include this amended Role of Funder statement in your cover letter; we will change the online submission form on your behalf. 4. Thank you for stating the following in the Acknowledgments Section of your manuscript: We are grateful to all study participants, RAPTOR-C19 site staff, and the staff of the NIHR Clinical Research Network, Thames Valley and South Midlands for their support for the study. We would like to thank Micheal McKenna for RedCAP support, patients and practices in the Oxford-Royal College of General Practitioners Research and Surveillance Centre (RSC) who share pseudonymised data to support research and surveillance (UKHSA is the principal sponsor of the RSC); EMIS, TPP, Vision and Wellbeing for assistance with pseudonymised data extraction. Membership of the RAPTOR-C19 Study Group is as follows: Katie Arundell1, Gail N. Hayward1, F. D. Richard Hobbs, Heather Kenyon, Joseph J. Lee, Kathryn Lucas, Brian D. Nicholson, Jessica Smylie, Sharon Tonner, Philip J. Turner.Nuffield Department of Primary Care Health Sciences, University of Oxford, UKNIHR Clinical Research Network, Thames Valley and South Midlands, Oxford, UK We note that you have provided funding information that is not currently declared in your Funding Statement. However, funding information should not appear in the Acknowledgments section or other areas of your manuscript. We will only publish funding information present in the Funding Statement section of the online submission form. Please remove any funding-related text from the manuscript and let us know how you would like to update your Funding Statement. Currently, your Funding Statement reads as follows: University of Oxford Medical Sciences Division Benefactors Urgent COVID-19 Fund [COVID-19 Research Response Fund Grant 0009325] to BDN, FDRH, JJL, TRF, PJT, GNH, MZ; National Institute for Health and Care Research (NIHR) School for Primary Care Research [SPCR grant 495] to BDN, FDRH, GNH, PJT, JJL, TRF; Urgent Public Health funding received by the CONDOR platform from the NIHR and Asthma + Lung UK [NIHR UPH grant COV0051] to BDN, PJT, GNH. PT, TRF, MG, UU, FDRH and GNH have received funding from the National Institute for Health and Care Research (NIHR) Community Healthcare MedTech and In Vitro Diagnostics Co-operative (MIC) (MIC 2016-018). PJT, TRF, MG, UU, BN and GNH currently receive funding from the NIHR HealthTech Research Centre (HRC) in Community Healthcare (NIHR205287) at Oxford Health NHS Foundation Trust. TRF and FDRH receive funding from the NIHR Applied Research Collaboration Oxford and Thames Valley at Oxford Health NHS Foundation Trust (NIHR200172). The views expressed are those of the authors and not necessarily those of the NHS, the NIHR or the Department of Health and Social Care. Roche International Ltd provided consumables and site training free of charge, together with a grant to the University of Oxford to support the study. Please include your amended statements within your cover letter; we will change the online submission form on your behalf. 5. One of the noted authors is a group or consortium. In addition to naming the author group, please list the individual authors and affiliations within this group in the acknowledgments section of your manuscript. Please also indicate clearly a lead author for this group along with a contact email address. 6. Please include captions for your Supporting Information files at the end of your manuscript, and update any in-text citations to match accordingly. Please see our Supporting Information guidelines for more information: http://journals.plos.org/plosone/s/supporting-information.

Reviewers' comments:

Reviewer's Responses to Questions

**Comments to the Author**

1. Is the manuscript technically sound, and do the data support the conclusions?

Reviewer #1: Partly

2. Has the statistical analysis been performed appropriately and rigorously? 

Reviewer #1: Yes

3. Have the authors made all data underlying the findings in their manuscript fully available?

Reviewer #1: Yes

4. Is the manuscript presented in an intelligible fashion and written in standard English?

Reviewer #1: Yes

5. Review Comments to the Author

**Reviewer #1:**  In this manuscript authors have explored the diagnostic accuracy of a point-of-care antigen test for SARS-CoV-2 and influenza in primary care facilities. Overall, the study is well structured and data is nicely presented, however, few areas need improvement and clarification.

1. The POCT was conducted unilateral nasopharyngeal samples from participants while the reference test was conducted using combined swabs from nasal and oropharyngeal sites. How can these different methods of taking samples affect the test results? or can ensure a consistency in the method to compare these tests sensitivities? Previously, it has been reported in some studies that combined naso-oropharyngeal swab can be more sensitive as compared to nasopharyngeal swab (10.1080/23744235.2018.1546055, 10.1002/rmv.2106). So, it would be interesting to see if the samples for both tests are taken by the same procedure.

2. Check the values presented in the table 1 particularly for Previous COVID-19 infection (Positive antigen test reported) if the values are represented as number (%) as mentioned in the table heading.

6. PLOS authors have the option to publish the peer review history of their article (what does this mean? ). If published, this will include your full peer review and any attached files.

**Do you want your identity to be public for this peer review?** For information about this choice, including consent withdrawal, please see our Privacy Policy .

Reviewer #1: No

---

## [Author Response · Author response to Decision Letter 1]

9 May 2025

We have addressed the editorial and reviewer comments below:

1. We have revised any formatting errors and apologies that these were missed on initial submission.

2. We have updated the financial information and the disclosure (see below).

3. Please include our revised statement - University of Oxford Medical Sciences Division Benefactors Urgent COVID-19 Fund [COVID-19 Research Response Fund Grant 0009325] to BDN, FDRH, JJL, TRF, PJT, GNH, MZ; National Institute for Health and Care Research (NIHR) School for Primary Care Research [SPCR grant 495] to BDN, FDRH, GNH, PJT, JJL, TRF; Urgent Public Health funding received by the CONDOR platform from the NIHR and Asthma + Lung UK [NIHR UPH grant COV0051] to BDN, PJT, GNH. PT, TRF, MG, UU, FDRH and GNH have received funding from the National Institute for Health and Care Research (NIHR) Community Healthcare MedTech and In Vitro Diagnostics Co-operative (MIC) (MIC 2016-018). PJT, TRF, MG, UU, BN and GNH currently receive funding from the NIHR HealthTech Research Centre (HRC) in Community Healthcare (NIHR205287) at Oxford Health NHS Foundation Trust. TRF and FDRH receive funding from the NIHR Applied Research Collaboration Oxford and Thames Valley at Oxford Health NHS Foundation Trust (NIHR200172). The views expressed are those of the authors and not necessarily those of the NHS, the NIHR or the Department of Health and Social Care. Roche International Ltd (R69261/CN085) provided consumables and site training free of charge, together with a grant to the University of Oxford to support the study. The funders had no role in study design, data collection and analysis, decision to publish, or preparation of the manuscript.

4. We would like to clarify that no one thanked in the acknowledgements provided direct funding but supported as part of existing national research infrastructure.

5. The required information is now included in the acknowledgements section of the manuscript.

6. We have included captions for the supplementary information files.

Reviewer 1 commented on the use of nasopharyngeal swabs for the POCT and a combined nasal/oropharyngeal swab for the reference test. Selection of upper respiratory tract swabbing sites was guided by manufacturers’ instructions, and while the literature shows some contrasting results in the direct comparison of these sites, evidence from systematic review suggests that they usually perform similarly for RT-PCR, so we think it unlikely that this would have had a major impact on our results. We have added a sentence to the Discussion describing this.

Reviewer 1 also asked us to check the information presented in Table 1. We have reviewed/corrected this table and apologise that one row pointed out by the reviewer was included in error; this has now been deleted.

---

## [Editor Report · Decision Letter 1]

21 Jul 2025

Diagnostic accuracy evaluation of a point-of-care antigen test for SARS-CoV-2 and influenza in UK primary care (RAPTOR-C19)

PONE-D-24-60482R1

Dear Dr. Tonner,

We’re pleased to inform you that your manuscript has been judged scientifically suitable for publication and will be formally accepted for publication once it meets all outstanding technical requirements.

Kind regards,

Yury E Khudyakov, PhD

Academic Editor

PLOS ONE
---

## [Editor Report · Acceptance letter]

PONE-D-24-60482R1

PLOS ONE

Dear Dr. Tonner,

I'm pleased to inform you that your manuscript has been deemed suitable for publication in PLOS ONE. Congratulations! Your manuscript is now being handed over to our production team.

Kind regards,

on behalf of

Dr. Yury E Khudyakov

Academic Editor

PLOS ONE